# Trypanosoma brucei triggers a broad immune response in the adipose tissue

**Henrique Machado**[1]☯, **Tiago Bizarra-Rebelo**[1]☯, **Mariana Costa-Sequeira**[1], **Sandra Trindade**[1], **Tânia Carvalho**[1], **Filipa Rijo-Ferreira**[2], **Barbara Rentroia-Pacheco**[1], **Karine Serre**[1]‡*, **Luisa M. Figueiredo**[1]‡*

**1** Instituto de Medicina Molecular João Lobo Antunes, Faculdade de Medicina, Universidade de Lisboa, Lisbon, Portugal, **2** Department of Neuroscience, Peter O'Donnell Jr. Brain Institute, University of Texas Southwestern Medical Center, Dallas, Texas, United States

☯ These authors contributed equally to this work.
‡ KS and LMF also contributed equally to this work.
* karineserre@medicina.ulisboa.pt (KS); lmf@medicina.ulisboa.pt (LMF)

## Abstract

Adipose tissue is one of the major reservoirs of *Trypanosoma brucei* parasites, the causative agent of sleeping sickness, a fatal disease in humans. In mice, the gonadal adipose tissue (AT) typically harbors 2–5 million parasites, while most solid organs show 10 to 100-fold fewer parasites. In this study, we tested whether the AT environment responds immunologically to the presence of the parasite. Transcriptome analysis of *T. brucei* infected adipose tissue revealed that most upregulated host genes are involved in inflammation and immune cell functions. Histochemistry and flow cytometry confirmed an increasingly higher number of infiltrated macrophages, neutrophils and CD4+ and CD8+ T lymphocytes upon infection. A large proportion of these lymphocytes effectively produce the type 1 effector cytokines, IFN-γ and TNF-α. Additionally, the adipose tissue showed accumulation of antigen-specific IgM and IgG antibodies as infection progressed. Mice lacking T and/or B cells (*Rag2*⁻/⁻, *Jht*⁻/⁻), or the signature cytokine (*Ifng*⁻/⁻) displayed a higher parasite load both in circulation and in the AT, demonstrating the key role of the adaptive immune system in both compartments. Interestingly, infections of *C3*⁻/⁻ mice showed that while complement system is dispensable to control parasite load in the blood, it is necessary in the AT and other solid tissues. We conclude that *T. brucei* infection triggers a broad and robust immune response in the AT, which requires the complement system to locally reduce parasite burden.

## Author summary

African trypanosomiasis is a neglected disease with significant socio-economic burden in sub-Saharan Africa. The protozoan parasite *Trypanosoma brucei*, a causative agent of African trypanosomiasis, can be found in the blood and extra-vascular spaces of the infected host. For an unknown reason, *T. brucei* accumulates in adipose tissue (AT) in very high numbers. Here we used a multidisciplinary approach to assess whether an immune response was mounted in AT during a *T. brucei* infection. We found that as

**Data Availability Statement:** RNA-Seq data is publicly available in the ArrayExpress database under the accession number E-MTAB-4061 and E-MTAB-7596.

**Funding:** This work was supported by the European Research Council (FatTryp, ref. 771714) awarded to LMF, by Fundação para a Ciência e Tecnologia (CEECIND/03322/2018) awarded to LMF, (PTDC/MED-IMU/30948/2017 and CEECIND/ 00697/2018) awarded to KS, (PD/BD/128286/ 2017) awarded to HM, (SFRH/BPD/89833/2012) awarded to ST, (IMM/BI/83-2017 through PTDC/ BIM-MET/4471/2014) awarded to TB-R and by the National Institutes of Health (NIGMS K99GM132557) awarded to FR-F. The funders had no role in study design, data collection and analysis, decision to publish, or preparation of the manuscript.

**Competing interests:** The authors have declared that no competing interests exist.

infection progresses, a broad variety of immune cells and antibodies accumulate in the AT. We also found that this broad immune response is partially able to control parasite numbers in the AT. Our study provides evidence that *T. brucei* parasites present in the AT are subjected to immune surveillance. The reason why *T. brucei* accumulates to such a high extent in AT remains to be elucidated.

## Introduction

*Trypanosoma brucei* is an extracellular protozoan parasite that causes sleeping sickness in humans and nagana in cattle, diseases that still hold a significant socio-economic impact in sub-Saharan Africa[1, 2]. *T. brucei* transmission to a mammalian host occurs upon the bite of an infected tsetse fly (*Glossina* spp). During such blood meal, parasites are released into the circulation and quickly differentiate into bloodstream forms (BSFs) that proliferate and invade the interstitial spaces of many organs[3]. In primate hosts, most African trypanosome species are rapidly eliminated by an innate immune trypanosome lytic factor (TLF)[4, 5]. This TLF is delivered by germline-encoded antibodies[6] and promotes complete parasite elimination before the onset of disease.

In non-primate mammalian hosts, or in primates infected TLF-resistant *T. brucei*, most parasite elimination does not occur through direct humoral mediated lysis and instead requires internalization by the host's phagocytes, mainly monocytes and macrophages. Optimal phagocytosis of *T. brucei* requires antibody and complement mediated opsonization[7]. These antibodies are produced by B cells, particularly plasmocytes (plasmablasts and plasma cells), activated during infection and are directed primarily at the parasite's variant surface glycoprotein (VSG)[7, 8]. This allows for direct Fc receptor-mediated phagocytosis and for the classical activation of the complement system, followed by phagocytosis.

Activation of both macrophages/monocytes and B cells is promoted by a T cell response comprising CD4+ T cells and CD8+ T cells[9]. Macrophages/monocytes respond to cytokines produced by T cells, such as interferon gamma (IFN-γ) and tumour necrosis factor alpha (TNF-α), by increasing phagocytic capacity, phagosome acidification and production of microbicidal reactive nitrogen species[10, 11]. Additionally, CD4+ T cells play an important role in B cell survival and maturation, which allows for the production of higher affinity antibodies via B cell class switching and affinity maturation[12]. Although this immune response is capable of largely limiting the number of parasites, it does so at the cost of severe immunopathology. To limit this deleterious effect, the immune system partially downregulates itself through the production of anti-inflammatory cytokines such as interleukin (IL)-10 which is highly expressed by regulatory T (Treg) cells[13, 14].

These host-parasite interactions, together with the capacity of parasites to switch VSGs by antigenic variation[15] and to undergo terminal differentiation to transmissible forms[16], leads to the characteristic oscillating waves of parasitemia displayed throughout the disease[17].

Most studies on the immune response against *T. brucei* have focused on blood and lymphoid organs. However, *T. brucei* occupies several extra-vascular tissues, including the brain, skin[18], testis[19, 20] and adipose tissue (AT)[21], which may present distinct immune responses[22] and even be immune privileged sites. A study from our laboratory showed that the AT is not only one of the major parasite reservoirs, reaching levels comparable to the blood, but also that parasites in this tissue (adipose tissue forms–ATFs) present a gene expression signature different from bloodstream forms[21].

For a long time, the role of the AT was believed to be exclusively of lipid storage, exerting important roles in energy homeostasis and thermogenesis[23]. It is now recognized that the AT is an immunologically active[24] endocrine organ[25], as adipocytes secrete a wide variety of immunomodulatory molecules[26]. *T. brucei* is not the only parasite that accumulates in the AT. *Neospora caninum*[27], *Plasmodium berghei* and *Trypanosoma cruzi* also reside in this tissue[28].Some protozoan parasites can even challenge the anti-inflammatory balance in AT or take advantage of such an environment to establish a successful infection. For instance, upon infection, *N. caninum* was reported to trigger a shift towards a pro-inflammatory environment, promoting pathogen elimination from AT[27]. On the contrary, *T. cruzi* was described to induce an anti-inflammatory macrophage polarization in a diet-induced obesity model, contributing to tissue homeostasis and parasite survival[29].

Given the importance of tissue immunity in a wide variety of conditions such as in infections[30] and cancer[31], here we compared the circulating, splenic and AT type of immune responses mounted against *T. brucei*. We show that the AT is a favorable location where an effective protective immune response is mounted, with both innate and adaptive immune cells that control parasite load in this tissue. We conclude that like in the blood, in the AT parasites require active mechanisms of immune evasion.

## Results

### Transcriptome of infected adipose tissue reveals a strong inflammatory response

A local immune response typically consists in the accumulation of myeloid and lymphoid cells sensing determinants of inflammation and/or determinants specific to the pathogen in order to control the infection and regulate the immune response itself. Given that the signature profile of immune cells is unique and different from all other resident cells of AT (adipocytes, endothelial cells, and mesenchymal stromal cells)[32, 33], we postulated that a transcriptome analysis of infected versus non-infected AT should provide a first glimpse of the type of immune response that is mounted in this tissue. We performed RNA sequencing (RNA-Seq) analysis of mice gonadal AT depots at early and late stages of infection. Total RNA was extracted from this tissue at day 0 (n = 3), 6 (n = 3) and 26 (n = 2) post-infection. Most sequence reads mapped to the mouse genome (between 86% and 96%) and the 1%-10% of reads that mapped to the parasite were not considered in this analysis (S1 Table).

Unbiased clustering of the expression profiles of the analysed samples showed that non-infected and infected AT clustered separately. Furthermore, the cluster of infected AT was divided in 2 sub-clusters separating samples of early and late infection time points (Fig 1A). This indicates significant alterations of transcript abundances in AT upon and during a *T. brucei* infection. To identify the genes differentially expressed, we used three distinct algorithms. Genes significant in at least 2 of them (adjusted *p*-value < 0.01) and having a fold-change higher than 2 were considered differentially expressed. At day 6 post-infection, 2678 genes were differentially expressed compared with non-infected AT. From these, 1770 genes were upregulated while the remaining 908 were downregulated at day 6 post-infection (S1C File). When comparing AT at day 26 post-infection with non-infected AT, 3684 genes were differentially expressed, from which 2264 were upregulated and 1420 downregulated (S1D File). Between AT at day 6 and day 26 post-infection, we identified 941 differentially expressed genes, 503 upregulated and 438 downregulated at day 26 post-infection (S1E File).

Gene ontology (GO) enrichment analysis was conducted to determine which GO terms were over-represented among the differentially expressed genes (Fisher's exact test, *p*-value < 0.01). Enrichment tests on the upregulated genes in infected AT (both on day 6

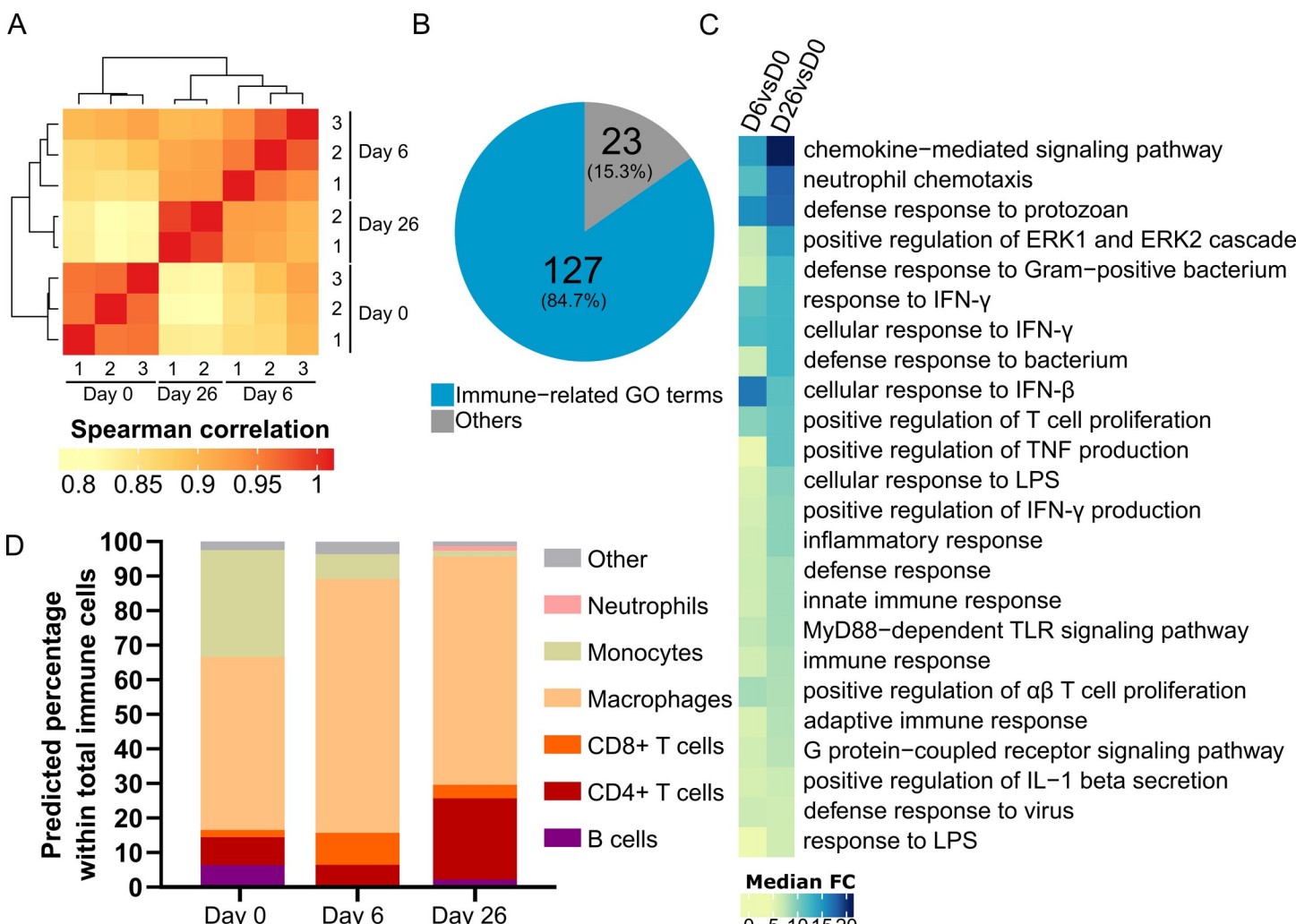

**Fig 1. Transcriptome of *T. brucei* infected AT shows a strong immune response. (A)** Heat map of hierarchical clustering of Spearman correlations of Reads per kilobase per million mapped reads (RPKM) levels from non-infected (D0), n = 3 and infected AT at early (D6), n = 3 and late (D26), n = 2 time points. **(B)** Pie chart of most significant biological processes GO term families. **(C)** Heat map view of median fold change (FC) of genes associated to the top 20 Biological Process GO terms in day 6 and day 26 of infection versus non-infected AT (D6 vs D0 and D26 vs D0, respectively) **(D)** Prediction of immune cell distribution based on immuCC RNA-seq deconvolution.

and day 26 versus day 0) revealed a strong inflammatory profile, with 84.7% of the top 150 (127/150) most significant Biological Process GO-terms related to immunity (Fig 1B and S2 File). Also, the median fold-change of the genes associated to the top 20 Biological Process GO-terms in infected versus non-infected AT (*p*-value < 1x10$^{-5}$) increased from day 6 to day 26 (Fig 1C), suggesting that the immune response taking place in AT increases in magnitude as infection progresses. Conversely, most downregulated genes were involved in metabolite biosynthesis (*e.g.* fatty acid biosynthetic processes) and other metabolic changes (*e.g.* fatty acid beta-oxidation) (S1 and S2D and S2E Files). Interestingly, differential expression analysis showed a marked and significant upregulation of T helper 1 (Th1) signature genes (*e.g. Tbx21*, *Eomes*, *Il12r1b*, *Ifng*, *Tnf*, *Nos2*, *Stat1*, *Stat4)* both on day 6 and day 26 versus day 0 (S1A Fig and S1C and S1D File), which are associated with a protective immune response against *T. brucei*. This profile was not observed for non-protective Th2 response associated genes (*e.g. Gata3*, *Il4*, *Il5*, *Il13*, *Stat5*, *Stat6)*, which were either not detected or

not differentially expressed in all conditions assessed (S1B Fig and S1 File). In addition to Th1 signature genes, the infected AT showed an overall upregulation of pro-inflammatory genes such as *Gzma*, *Gzmb*, *Il1a and Il6* (S1A Fig). In turn, this was accompanied by the upregulation of genes such as *Il10*, *Il10ra*, *Ctla4*, *Foxp3* and *Pdcd1* which are associated with the suppression of an active immune response (S1B Fig).

To assess whether the changes in transcriptomic profile of the AT could be due to an alteration of the cellular makeover of the tissue, we performed a cellular deconvolution analysis using the immuCC algorithm[34]. This analysis predicted that during a *T. brucei* infection the relative proportions of innate and adaptive immune cells were variable (Fig 1D). Specifically, these data predicted a relative increase of macrophages and CD8+ T cells between days 0 and 6 post-infection, while the prediction for monocytes and B cells was a relative decrease in the same period. Lastly, by day 26 post-infection this deconvolution model predicted a sizeable increase of the CD4+ T cell response, with a predicted 3 to 4-fold relative increase when compared to days 0 and 6 post-infection. Overall, the transcriptome of the AT of infected mice revealed signs of intense inflammation and an ongoing active immune response.

## Adipose tissue shows a gradual accumulation of immune cells during infection

Bulk RNA-seq provides a strong indication of broad changes in the inflammatory profile immune cell milieu of the AT, however it does not allow to identify the absolute accumulation of immune cells and determine their selective *in situ* effector functions. To investigate this, and to compare the extent of systemic to AT immune responses, we quantified the number of parasites and immune cells in the spleen and the AT. The spleen was used as a proxy for the systemic immune response as it filters bloodborne pathogens and is a major site for lymphocyte activation and proliferation. Quantifications were performed at key time-points of the murine *T. brucei* infection, encompassing the formation and resolution of the first and second peaks of parasitemia as well as the chronic stage of infection.

In the blood, the progression of parasitemia throughout infection exhibited its characteristic pattern (Fig 2A): a first peak around day 5 post-infection; undetectable parasitemia between days 9 and 13 post-infection; followed by a fluctuating number of parasites that differs in each mouse until day 28 post-infection. In the spleen, the number of parasites reached the peak at day 5 post-infection and dropped 32-fold from 6 to 9 days post-infection, when the lowest parasite load was detected (Fig 2B). From day 14 post-infection, the number of parasites fluctuated at around $10^5$ parasites per organ. Remarkably, in the gonadal AT, the number of parasites peaked at day 6 post-infection (one day later than in blood/spleen) and it suffered only a 6-fold reduction from 6 to 9 days post-infection (Fig 2B). From day 14 post-infection onward, the number of parasites fluctuated at around $10^6$ parasites, which is an approximate constant 10-fold more than in the spleen or any other organ assessed, as we have previously reported[21].

At the same key time-points of infection, immune cells from spleen and AT were analysed by flow cytometry. Using an anti-CD45 antibody to gate all immune cells (S2 Fig), we observed a striking gradual accumulation in the gonadal AT with a 16-fold increase in the number of immune cells per tissue from days 0 to 28 post-infection (Fig 2C). Conversely, and despite a large increase in spleen mass (S4A Fig), no significant changes were observed in the total number of splenic immune cells. This suggests that the increase in spleen weight may be largely due to the accumulation of damaged red blood cells undergoing eythrophagocytosis[35], which is known to occur during *T. brucei*-induced anemia[36].

Interestingly, while the number of parasites in the AT peaks on day 6 post-infection and oscillates afterwards, the immune cells are recruited and/or proliferate in AT with different dynamics,

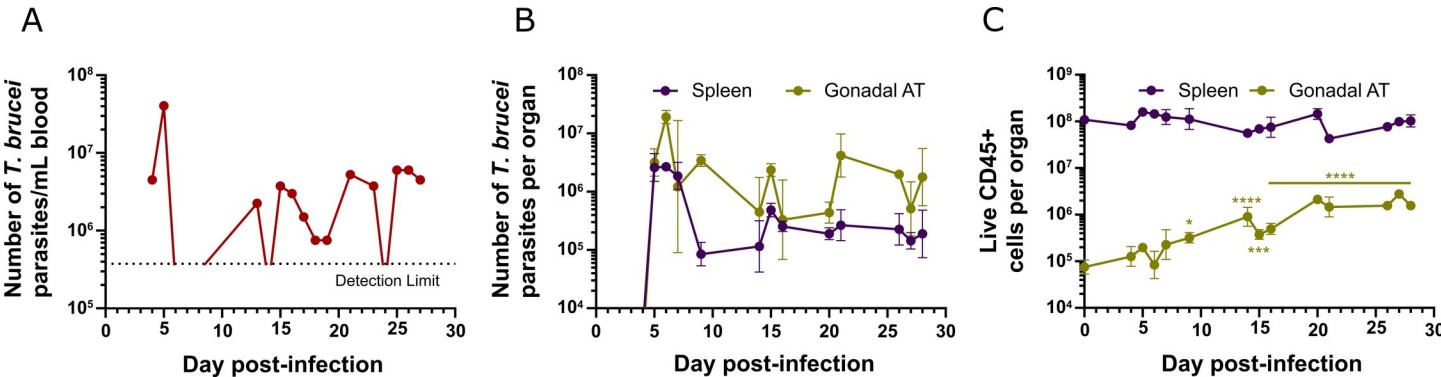

**Fig 2. Dynamics of *T. brucei* parasites and immune cells throughout infection.** (A) Number of *T. brucei* parasites per mL of blood of a single representative mouse, quantified using a hemocytometer. Dashed line represents the detection limit ($3.75 \times 10^5$ parasites/mL of blood). (B) Total number of *T. brucei* parasites in the spleen and gonadal AT, quantified by qPCR. (C) Number of live immune cells in spleen and gonadal AT. (B-C) Error bars represent the standard error of the mean (SEM) (n = 2–6 mice per group). Statistical analysis was performed with a two-way analysis of variance (ANOVA) using Sidak's test for multiple comparisons. * refers to statistical differences within the groups in each time-point and the non-infected group. *, $P < 0.05$; **, $P < 0.01$; ***, $P < 0.001$; ****, $P < 0.0001$.

increasing steadily as infection progresses. The fact that the spleen has a higher proportion of immune cells than the AT is not surprising due to its nature as a secondary lymphoid organ.

To assess the distribution of the inflammatory cell infiltrates and the morphological changes of the AT, we performed immunohistochemistry for parasites (anti-VSG antibody), macrophages (anti-F4/80 antibody) and T cells (anti-CD3 antibody) in sections of gonadal AT on days 6 and 26 post-infection. Consistent with the analysis by flow cytometry, a significant increase was observed in the number of infiltrating macrophages and T cells, mainly during the later time-points of infection (Fig 3A). Inflammatory cells were distributed diffusely in the tissue, often associated with parasites or parasite debris. By transmission electron microscopy (TEM), parasites were also detected intracellularly, in the cytoplasm of phagocytes (most likely macrophages) (Fig 3B). These intracellular parasites were often surrounded by membranous whorls, suggesting they had been phagocytosed.

In conclusion, our analysis shows that the AT is not an immune silent tissue, but instead that it actively responds to a *T. brucei* infection. It is important to mention that these data were acquired in an intraperitoneal infection model and that Tsetse fly transmitted or intradermally injected *T. brucei* may lead to significant differences in immune response.

## Adipose tissue is populated by effector innate and adaptive immune cells

To experimentally characterize the immune response that is mounted against *T. brucei* in the AT, we isolated gonadal fat pads and spleen of infected animals at different time-points of the infection, and we identified and quantified the main immune cell subsets by flow cytometry. On the one hand we followed the dynamics of neutrophils, monocytes, and macrophages to assess the innate myeloid immunity branch. On the other hand, to assess the adaptive immunity branch, we followed T cell subsets (helper T cells (CD4+) and killer T cells (CD8+)) and evaluated whether these T cells were activated and responding to the infection by assessing their expression of two main pro-inflammatory cytokines, IFN-γ and TNF-α. Additionally, we analysed the frequency of Treg cells, a key immunosuppressive subset. The gating strategies for all the populations are presented in the S2 Fig.

As infection progressed, while the total number of immune cells of all subtypes assessed increased in the AT, the same was not observed for the spleen (Fig 4). As expected for the innate immune response, neutrophils (Fig 4A) and monocytes (Fig 4B) are the first to be

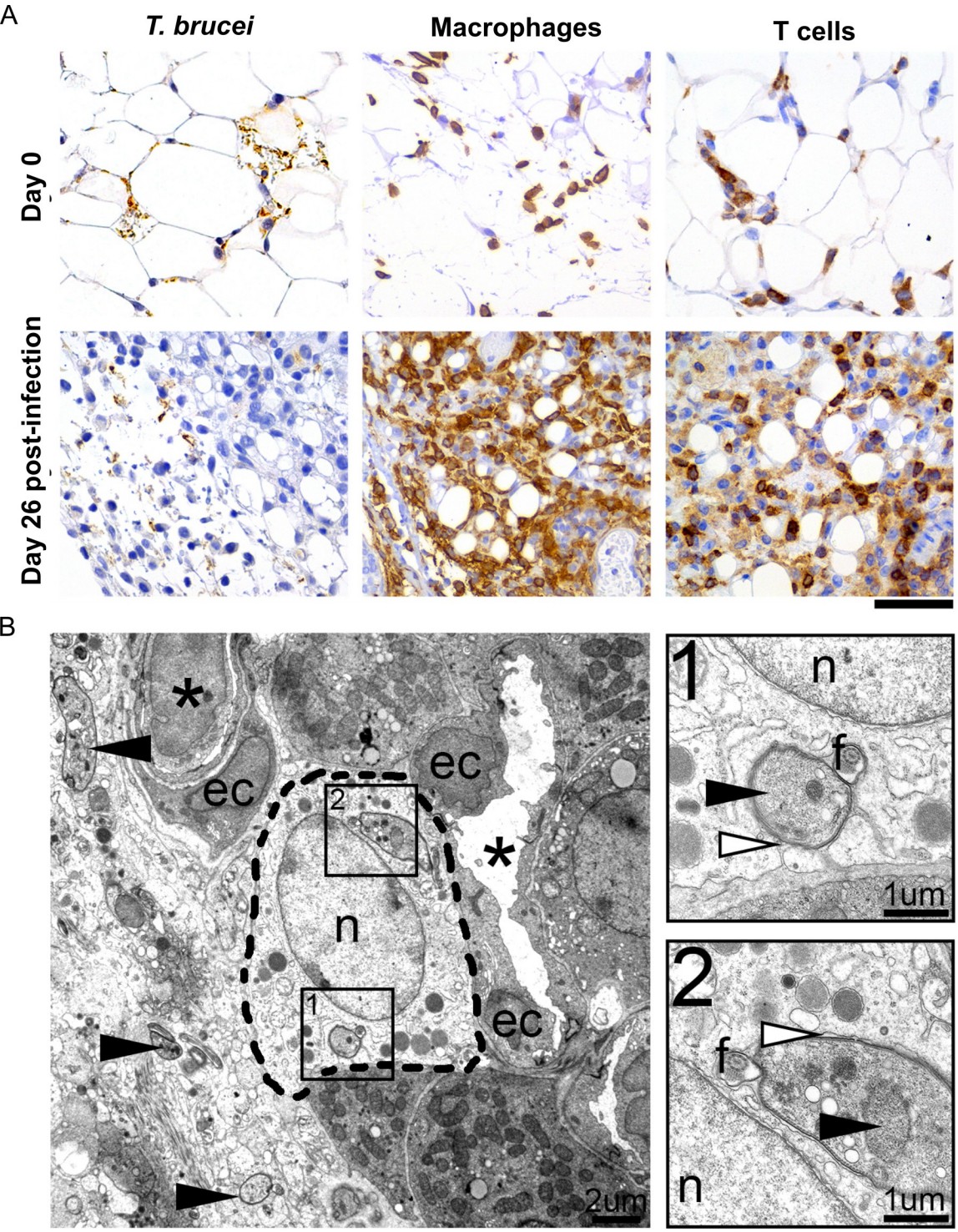

**Fig 3. Inflammatory cell response in the adipose tissue. (A)** Immunohistochemistry reveals a large number of *T. brucei* parasites (anti-VSG antibody) accompanied by marked infiltration by macrophages (anti-F4/80 antibody) and moderate infiltration by T cells (anti-CD3 antibody), at a later stage post-infection (day 26). (n = 4–6 per time-point; DAB counterstained with hematoxylin, original magnification 40x (Scale bar, 50μm). **(B).** Representative electron micrograph of the perirenal adipose tissue of the mouse, 26 days post-infection, showing various extracellular tangential and cross-sectional profiles of trypanosomes (arrowhead) and also intracellular parasites, in the cytoplasm of a phagocyte [most probably a macrophage (inside dashed line)]. Asterisk, vessel; ec, endothelial cell; n, nucleus, f, flagellum. **B-1, B-2.** Insets of the phagocytized trypanosomes, with well-defined nucleus (black arrowhead) and pseudopodia (white arrowhead), consisting of multiple membranous whorls extended by the phagocyte around the parasite.

recruited to the AT, presenting an increase of 68-fold and 49-fold from days 6 to 9 post-infection, respectively. Few neutrophils were detected in the gonadal AT prior to infection with $8.45 \times 10^2$ cells, however they increased nearly 100-fold by day 28 post-infection (Fig 4A). Similarly, the number of neutrophils in the spleen increased significantly, presenting a 38-fold increase from days 0 to 28 post-infection (Fig 4A).

Prior to infection, few monocytes were present in the spleen and AT. During *T. brucei* infection, monocytes accumulated in both tissues, increasing 39-fold in the spleen and 12-fold in the gonadal AT by day 28 post-infection (Fig 4B).

Macrophages were the most abundant myeloid population in both tissues in control mice. Interestingly, this population was kept at relatively stable numbers throughout infection in the spleen, as the only significant change observed was a moderate 2.7-fold increase by day 28 post-infection (Fig 4C). In the gonadal AT, the number of macrophages was significantly increased from day 20 post-infection onward, presenting a 10-fold increase by day 28 post-infection (Fig 4C).

CD4+ and CD8+ T cell subsets showed a similar overall increase in the AT, while the spleen showed no changes in CD4+ T cells and an overall decrease for CD8+ T cells (S3B and S3C Fig). Within the total pool of T cells, we assessed the fraction that differentiated into effector T cells expressing IFN-γ, TNF-α or both pro-inflammatory cytokines. During the infection, we observed that the number of effector CD4+ T cells presented a modest incremental trend in the spleen, reaching a 3.8-fold increase by day 28 post-infection (Fig 4D). A higher relative increase in effector CD4+ T cells was observed in gonadal AT, with significant accumulation from day 14 post-infection onwards, showing a 14.6-fold increase by day 28 post-infection (Fig 4D). A distinct profile between the spleen and gonadal AT was observed for effector CD8 + T cells. There was a trend for a modest reduction in number of this effector cell in the spleen (Fig 4E). In contrast, a significant increase in the number of effector CD8+ T cells in the gonadal AT was observed from day 16 post-infection onwards, reaching a 8.7-fold increase by day 28 post-infection. This profile of effector T cells is in agreement with the high expression of TNF-α and IFN-γ in gonadal AT revealed at the transcriptomic level (S1 Fig and S1C and S1D File), confirming the accumulation of effector T cells in the AT and propagating a Th1 type of response. The number of Treg cells showed an overall decrease in the spleen while increasing 4-fold in gonadal AT (Fig 4F), which is in agreement with the increased expression of FOXP3, CTLA-4 and IL-10 at the transcriptomic level (S1 Fig and S1C and S1D File). This suggests that Treg cells recruited to AT may contribute to control an excessive inflammatory response and could favor the persistence of *T. brucei* parasites in AT.

This flow cytometry analysis in the AT is mostly consistent with changes observed in transcriptomic data (Fig 1C) and predicted by immuCC (Fig 1D). Specifically, we confirmed a similar relative variation for many immune subsets such as CD4+ T cells, CD8+ T cells, monocytes and neutrophils. Macrophages were the only population for which flow cytometry did not confirm the immuCC predictions (S2 Fig). This overestimation of the macrophage population by immuCC could be due to the fact that immune activation of adipocytes unlocks a gene expression signature that highly resembles that of macrophages[37]. Overall, during a *T. brucei* infection the AT accumulates cells of the innate and adaptive immunity branches. Importantly, there is a significant increase of immune cells described as protective against a *T. brucei* infection (*i.e.* IFN-γ+CD4+Th1 cells and macrophages).

## Adipose tissue presents a strong humoral response

Effective immunity against *T. brucei* requires a humoral immune response in addition to a strong cellular immune response[8]. The importance of antibody-mediated *T. brucei* clearance

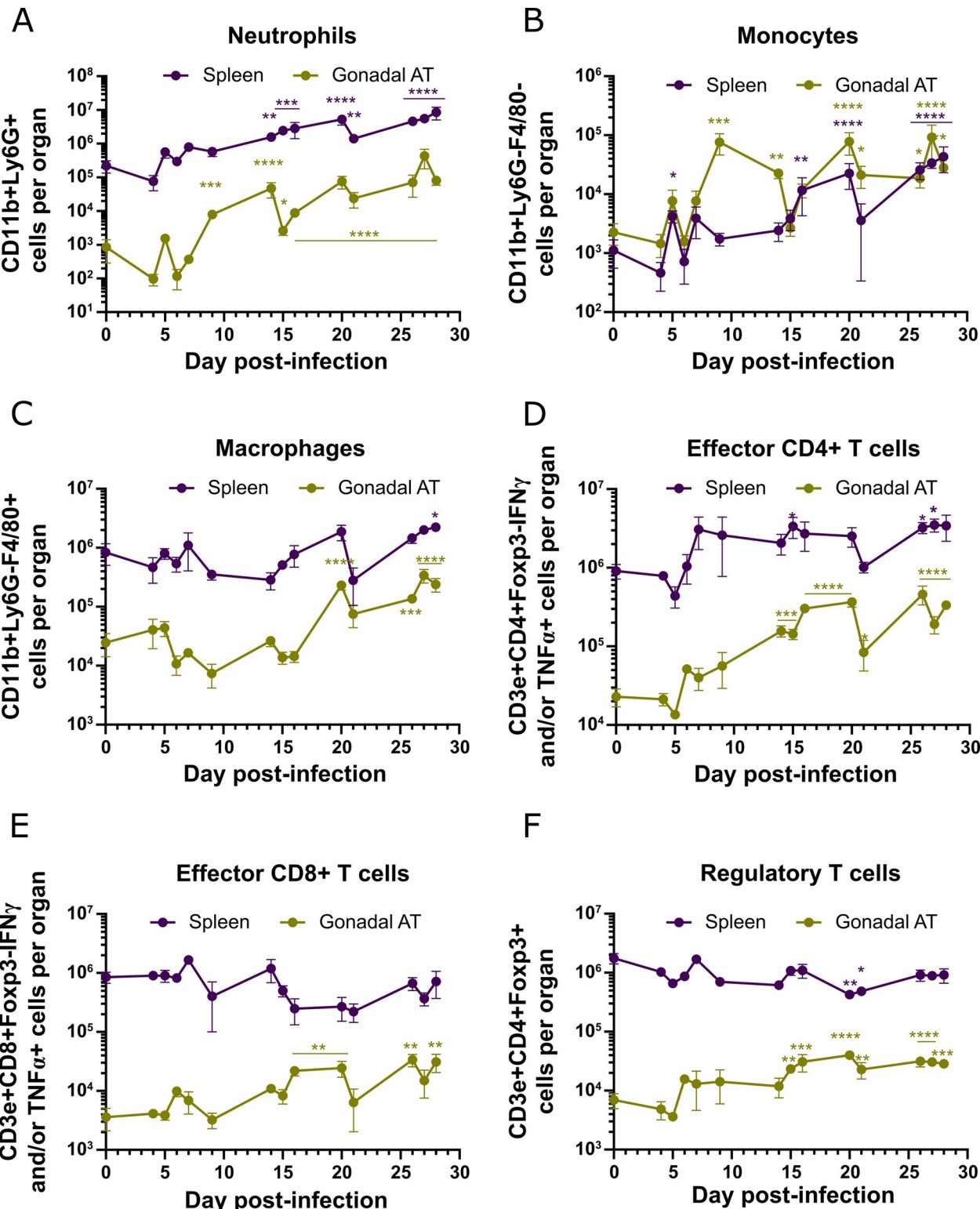

**Fig 4. Dynamics of immune cells during infection.** (A-F) Number of (A) neutrophils, (B) monocytes, (C) macrophages, (D) effector CD4+ T cells, (E) effector CD8+ T cells and (F) regulatory T cells in the spleen and gonadal AT. Error bars represent the SEM (n = 2–6 mice per group). Statistical analysis was performed with a two-way ANOVA using Sidak's test for multiple comparisons. * refers to statistical differences within the groups in each time-point and the non-infected group. *, P<0.05; **, P<0.01; ***, P<0.001; ****, P<0.0001.

is well documented. Indeed, B cell and antibody deficiencies lead to increased susceptibility to T. *brucei*[38] and passive transfer of VSG-specific antibodies grants variant-specific protection [39, 40].

To assess if a humoral response was ongoing in solid tissues and in AT in particular, we quantified the levels of antigen-specific IgM, IgG and IgA by ELISA, using plates coated with purified VSG AnTat1.1 (S5 Fig). A series of 10-fold dilutions of serum samples and 2-fold dilutions of cell-free suspensions from tissues were tested for anti-VSG AnTat1.1 immunoglobulins. For each sample, the antibody titer was determined as the highest dilution with an OD 450nm higher than the cutoff value (0.13x average of positive control + average of negative control) (S5D Fig)[41].

We collected samples from mice on days 0, 6 and 9 post-infection from the serum (reference of circulating immunoglobulins), spleen (reference of an active site of immunoglobulin production), AT and kidney (an example of low parasitised organ). These timepoints were chosen because in wild-type (WT) mice there is a significant drop in AT parasite burden between days 6–9 (Fig 2B) and we questioned if B cell responses could play a local role. As expected, circulating antibodies against VSG AnTaT1.1 increased in infected mice when compared to non-infected mice (*i.e.* 10 to 100-fold increase (Fig 5A). This effect was also observed in the interstitial fluid, albeit in an organ specific manner. Specifically, the antibody titer of IgM in the AT increased upon infection but was lower than in the kidney and spleen at day 6 post-infection (6.6 and 88-fold respectively) and to a lesser extent at day 9 post-infection (4.6 and 6.3-fold). Moreover, the antibody titer for IgG in the AT was similar that of the kidney at day 6 post-infection but was 107-fold lower than that of the spleen. By day 9 post-infection the AT presented a 3-fold lower anti-VSG IgG titer compared to the kidney and 14-fold lower than the spleen. Additionally, anti-VSG IgA was not detected in the AT and presented low titers in the spleen. Interestingly, kidney titers for anti-VSG IgA were high but were independent of infection, as titers from non-infected mice were equally elevated as those of infected mice.

Next, we assessed whether this humoral response in the AT was accompanied by a local B cell response. There was an increase in the numbers of B cells in spleen and AT (Fig 5E), as well as an increase in the number of activated, (IgD-CD19+ cells), and proliferating (Ki67+) B cells (Fig 5F). Specifically, by day 9 post-infection total B cells showed a 3.55-fold increase in the spleen and a 8.45-fold increase in the AT, relative to non-infected controls. This increase was also observed for activated B cells, as their number presented a 16.29-fold increase in the spleen and a 9.24-fold increase in the AT in the same period. We then assessed whether plasmocytes (*i.e.* main immunoglobulin producing cells) were present in the AT. CD138 (also known as syndecan-1) is an excellent marker of plasmacytic differentiation, however it is sensitive to the collagenase treatment used for flow cytometry analysis of AT digestion, and thus we applied immunohistochemistry to AT (Fig 5G–5I). Relative to non-infected mice, the number of CD138+ cells increased 3.7-fold and 17.4-fold at days 6 and 9 post-infection, respectively (Fig 5J), indicating that the AT harbors plasmocytes that secrete immunoglobulins.

Collectively, these data show that a humoral response takes place in the AT and suggest that the increase of immunoglobulin levels is likely the result of the infiltration of circulating immunoglobulins (serum), but also partially due to increased local production by antibody-producing plasmocytes in the AT. Altogether, we conclude that the components of a protective immune response against *T. brucei* are present in the AT.

## Parasite load in adipose tissue depends on systemic effectors

Parasites infiltrate and accumulate in the AT, even though a host-protective acute (that turns chronic) immune response is mounted in response to the infectious agent, as revealed by the

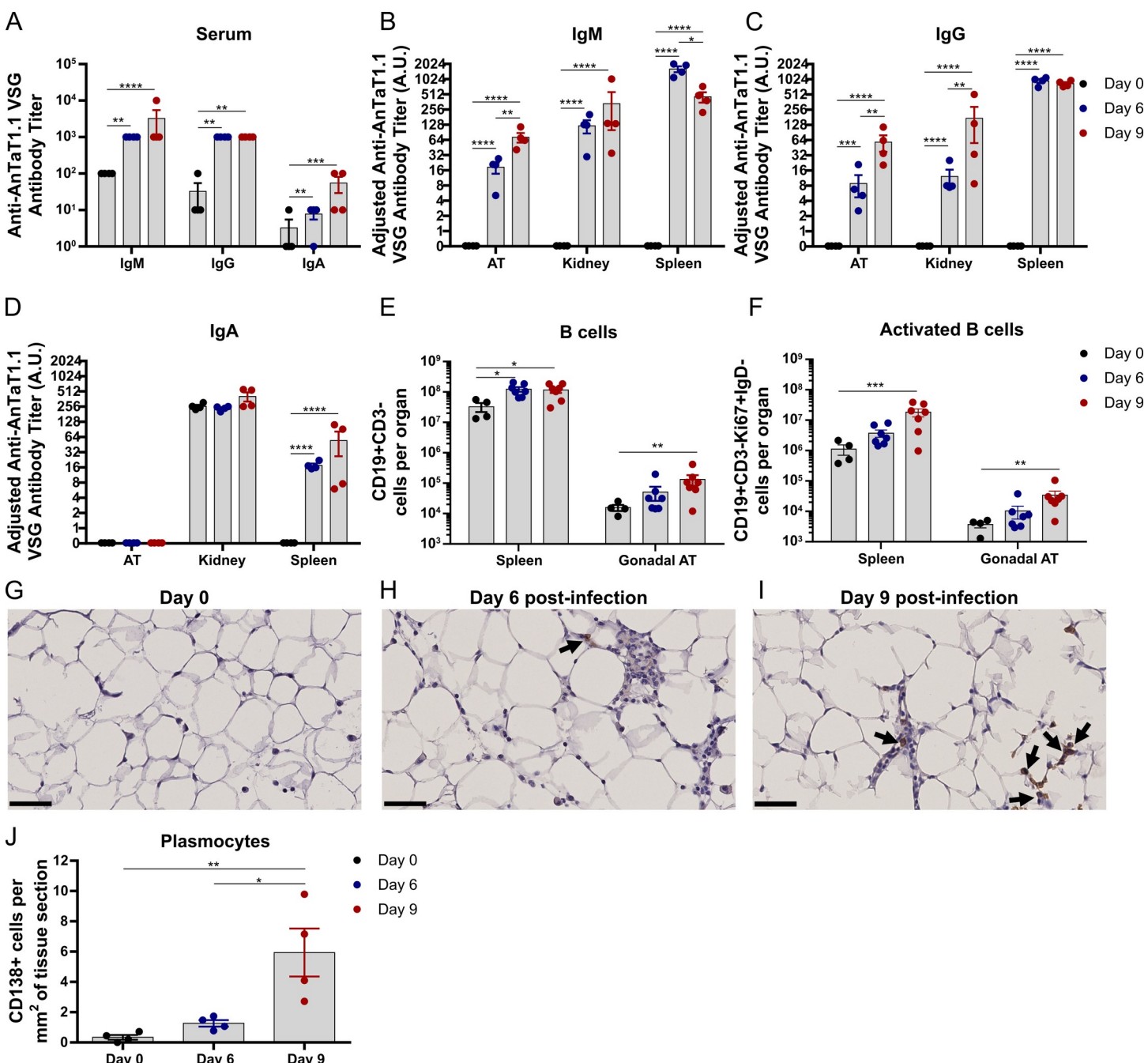

**Fig 5. Dynamics of immunoglobulins, B cells and plasmocytes during infection.** (A) Antibody titers of anti-AnTaT1.1 VSG in the serum. Antibody titer normalized to tissue mass used to prepare cell-free suspensions in arbitrary units (A.U.) of (B) IgM, (C) IgG and (D) IgA. Number of (E) total B cells and (F) activated B cells in the spleen and gonadal AT. Immunohistochemistry detection of (anti-CD138 antibody) plasmocytes at days (G) 0, (H) 6 and (I) 9 post-infection. (J) Quantification of plasmocytes in tissue sections. (G-I) Black arrows indicate CD138+ cells and scale bars represent 50 μm, original magnification 40x. Statistical analysis was performed with (A-F) two-way ANOVA and (J) one-way ANOVA using Sidak's test for multiple comparisons. *, P<0.05; **, P<0.01; ***, P<0.001; ****, P<0.0001.

dramatic accumulation of myeloid cells and lymphoid effector subsets (Figs 3 and 4). Thus, we decided to undertake a systematic approach and test the importance of several immune cells and soluble factors key in the control of *T. brucei*. For this, we performed infections in several immunocompromised mouse mutants, and collected organs on days 6 and 9 post-infection.

Specifically, we infected mice that lack T and B lymphocytes (deficient in the recombination activation gene 2 ($Rag2^{-/-}$)), (Fig 6A and 6B) mice that lack selectively B lymphocytes ($Jht^{-/-}$) (Fig 6C and 6D), mice that lack IFN-γ ($Ifng^{-/-}$) (Fig 6E and 6F), and mice that lack complement component 3 ($C3^{-/-}$) (Fig 6G and 6H). These cells or factors have been previously implicated in the parasite elimination. Here, we hypothesized that if one of these factors was ineffective solely in the clearance of parasites in the AT, its absence would have little impact on the elimination of AT parasites, but it would prevent parasite elimination in the blood and other organs.

Upon infection with *T. brucei*, in both $Rag2^{-/-}$ and $Jht^{-/-}$ mice, parasitemia increased until day 6 post-infection, but instead of undergoing a quick reduction, parasite numbers persisted at very high levels (Fig 6A and 6C). While WT mice were able to reduce parasite burden from the spleen and AT between days 6 and 9 by roughly 10-fold, neither $Rag2^{-/-}$ nor $Jht^{-/-}$ showed a significant decrease in the parasite burden of these organs in the same period (Fig 6B and 6D). These data show that efficient parasite elimination between days 6 and 9 post-infection in the AT requires a B and T cell responses, similarly to parasitemia clearance in the blood. Given that the infection promoted the generation of a high frequency of IFN-γ-producing CD4+ and CD8+ T effectors in AT (Fig 4D and 4E), the contribution of IFN-γ was assessed by infecting $Ifng^{-/-}$ mice. Unlike WT controls, $Ifng^{-/-}$ mice were not able to eliminate parasites from the blood nor from all tested solid tissues (Fig 6E and 6F). These results suggest that *T. brucei* is controlled by IFN-γ both systemically and within the tissues, highlighting the key role of Th1-dependent immune response in the control of the infection.

As previously shown, albeit with different kinetics[42], mice with defective complement system (C3-/-), were able to control the first peak of T. *brucei* parasitemia, suggesting that complement opsonization is not essential in the blood (Fig 6G). Interestingly, these mutant mice were partially impaired in controlling the number of parasites in the AT, between days 6 and 9 post-infection (4.3-fold reduction vs 32-fold reduction in WT mice) (Fig 6H). This impairment was also observed in the heart, where $C3^{-/-}$ showed virtually no parasite elimination (1.6-fold) whereas WT mice presented an 8.4-fold decrease between days 6 and 9 post-infection. These results suggest that the complement system may have a more active role in parasite elimination from some tissues than from the blood.

We then questioned whether an overarching immunosuppressive mechanism was reducing overall parasite clearance and allowing for higher parasite accumulation in the AT, when compared to other organs. Among these, the programmed cell death protein 1 (PD-1) and the programmed cell death ligand 1 (PD-L1) interactions are well described as inhibitors of T cell effector functions in several contexts, including infection[43]. The interaction between PD-1 on the surface of a T cell and PD-L1 on the surface of an antigen presenting cell, leads to the decreased activation, proliferation and cytokine secretion of the T cell. Although we observed by flow cytometry an upregulation of PD-1 in both CD4+ and CD8 + T cells (S6A and S6B Fig) and in *Pdcd1* gene expression (S1B Fig and S1C File) during the acute phase of infection, its antibody-mediated blockade failed to enhance parasite clearance in AT (S6C and S6D Fig).

Overall, the infections in immunocompromised mice revealed the parasite burden in the AT depends on several systemic components (B cells, T cells, IFN-γ and complement system). We conclude that the AT does not provide a niche devoid of immune surveillance for *T. brucei*.

## Discussion

The relevance of tissue invasion and colonization by *T. brucei* is only now being elucidated in the context of pathogenesis, persistence, and drug resistance. Each tissue presents a specific immune environment that may impact a pathogen's ability to colonize and persist in each

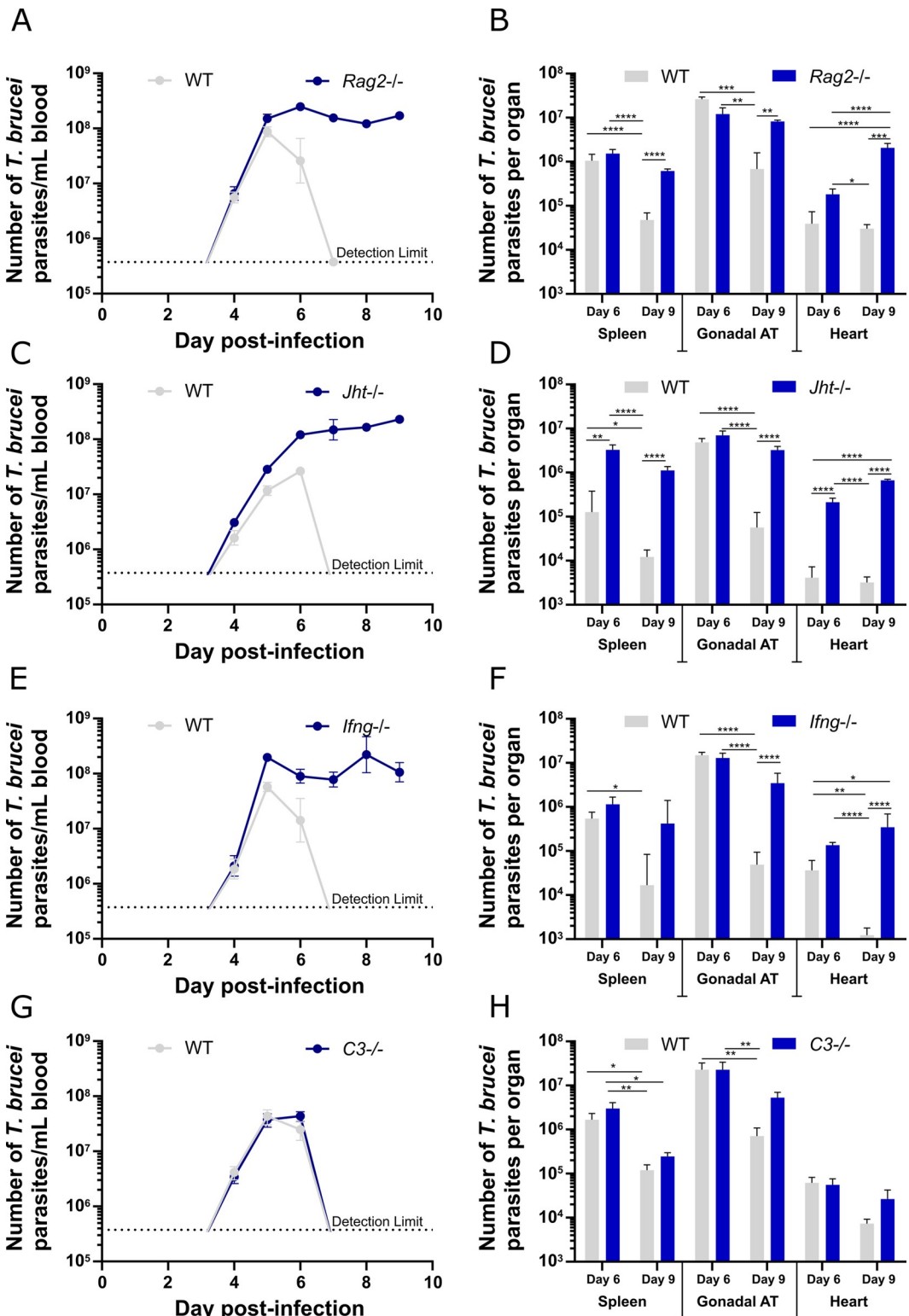

**Fig 6. Assessment of the role of immune cells/factors in parasite control.** Infection of WT and **(A-B)** *Rag2*[-/-], **(C-D)** *Jht*[-/-], **(E-F)** *Ifng*[-/-], **(G-H)** *C3*[-/-] **(A-H)** Error bars represent the SEM (n = 3–7 mice per group). **(Left)** Parasitemia of mice infected with *T. brucei*, quantified in a hemocytometer. Dashed line represents the limit of detection ($3.75 \times 10^5$ parasites/mL of blood). **(Right)** Number of *T. brucei* parasites quantified by qPCR in spleen, gonadal AT and heart, 6 and 9 days post-infection. Statistical analysis was performed with a two-way ANOVA using Sidak's test for multiple comparisons. *, $P < 0.05$; **, $P < 0.01$; ***, $P < 0.001$; ****, $P < 0.0001$.

anatomical location. The increasing evidence of persistence of evolutionarily diverse pathogens in the AT raises the question of its immune competence. Given the extensive accumulation of *T. brucei* in the AT we sought to elucidate whether an effective immune response was mounted in this tissue and how different it was from the better described systemic responses observed in the blood and spleen (where the parasites also reside).

## Adipose tissue immunity against *T. brucei*

Here, we show that upon *T. brucei* infection the AT acquires a marked immune transcriptomic phenotype. Within this immune phenotype, a marked Th1 signature is observed, which unlike a Th2 signature, is associated with protection against *T. brucei*. This shift in transcriptomic profile is likely the result of *in situ* activation of resident immune cells as well as of the recruitment of additional immune cells throughout infection. Accordingly, during infection we observed both by immunohistochemistry and flow cytometry a large accumulation of myeloid and lymphoid cells. Indeed, although with distinct kinetics, similar accumulations of neutrophils[44], monocytes[44, 45] and T cells[46, 47] have been reported for other active sites of *T. brucei*, such as the skin[44], liver[45, 47] and brain[46]. Comparison of the AT and spleen leukocyte accumulation kinetics reveals a sharp contrast between these tissues. The AT presents an overall sustained increase throughout time of all the immune cell types assessed.

While starting with a much higher number of immune cells, the spleen shows an overall maintenance of the total number of immune cells (Fig 2C). This maintenance of the total number of splenic immune cells does not represent a static immune environment and it is in fact the result of opposing variations in individual immune subsets. Among the few similarities found between the spleen and AT is the overall increase in monocyte and neutrophil numbers (Fig 4A and 4B), which is in accordance with previously published data for the spleen[48]. Although an active immune response is taking place in both the spleen and AT, the differences in their immune kinetics, especially in the T cell compartment, suggest that parasites in the spleen and in AT may be subjected to distinct immune responses.

The immune response mounted in the AT is likely to actively participate in the reduction of the number of parasites. Classically, parasites are described to be eliminated by antibody-dependent phagocytosis that can be assisted by IgM-mediated complement opsonization[9]. Our observation by TEM of phagocytosed parasites in the AT suggests that local elimination of the parasite is ongoing. This phagocytic clearance is consistent with the accumulation of antigen-specific IgM and IgG (Fig 5B and 5C) in the tissue, and macrophages that are activated by innate signals delivered by the presence of the parasite as well as adaptive signals such as the local production of IFN-γ by CD4+ Th1 cells (Fig 4D). This is supported by the observed abrogation of parasite clearance from the AT in infected mice deficient for T and/or B cells, or IFN-γ (Fig 6). Given the hyperparasitemia presented by these mutant mice, we cannot exclude the possibility that the high number of parasites in AT 9 days post-infection is partially due to a spill in effect from circulating parasites (although in some mutants and in some organs a tendency for higher parasite load is already detectable 6 days post-infection). However, the inability to remove immune cells/factors exclusively from the AT prevents the study of the contribution of circulating parasites that may excessively cross into tissues.

The dynamics of anti-VSG immunoglobulins show a protective humoral response in the AT, although at significantly lower levels than the spleen. Compared to the kidney, the AT shows comparable anti-VSG IgG titers and modestly lower IgM titers. Interestingly, while the AT shows no detectable anti-VSG IgA titer, the kidney has elevated titers even in non-infected mice. These are likely natural polyreactive IgAs [49] whose relevance in Trypanosomiasis, if any, remains to be elucidated. Although our data show the presence of plasmocytes and thus

suggest that local immunoglobulin production in the AT is likely occurring (Fig 5G–5J), how important this production is for the overall humoral response requires further investigation.

The observation that $C3^{-/-}$ mice are not as efficient at reducing parasite load in non-lymphoid solid tissues on day 9 is intriguing (Fig 6G and 6H). It has been previously reported that, while the parasite is at least partially resistant to complement-mediated lysis[50–52], the complement system is important for *T. brucei* opsonization and killing via phagocytosis[7, 52]. In the absence of complement opsonization, macrophages can still phagocytose *T. brucei*, albeit less efficiently[52–54]. Classical activation of the complement system requires IgM or IgG binding the parasite to initiate binding of complement components to the Fc-region. Unlike IgG, opsonization by IgM alone does not promote macrophage phagocytosis, as these cells lack the FcμR[55]. Hence, IgM-promoted phagocytosis requires opsonization by the complement system. Given that in the AT, anti-VSG IgM and IgG titers are significantly lower than in the spleen (a proxy for the blood) (Fig 5), it is tempting to speculate that the parasite control in AT (and perhaps other solid organs) is more dependent on complement opsonization, while in the spleen the faster accumulation of specific IgG could reduce complement-dependency. Further studies are required to better understand the role of the complement system in *T. brucei* infection, including a quantitative analysis of the complement components and immunoglobulins in multiple tissue types throughout infection.

Parasite immune evasion in the AT could also be achieved by an increased activity of immunosuppressive effectors when compared to other organs. Our transcriptomic data show an upregulation of multiple immune-suppressive genes (S1B Fig), which is further validated by a progressive accumulation of Treg cells (Fig 4F). This suppressive response serves to prevent overactivation of the immune system and reduce immunopathology but can be exploited by pathogens to reduce immune pressure on them. In this scenario it is possible that one or multiple of these suppressive mechanisms prevent sterile tissue immunity. As such, we sought to disrupt the immunosuppressive PD-1/PD-L1 axis which shows evidence of being upregulated in the AT during infection. This disruption led to no significant alterations in the number of parasites in the AT (S6D Fig). Thus, it is unlikely that this immunosuppressive axis is a determining factor for parasite colonization of the AT.

## How do parasites evade the immune response in the AT?

Antigenic variation is the hallmark immune evasion strategy used by African trypanosomes. At any given time during infection a small subset of VSGs is dominant among the BSF population[56], leading to the mounting of VSG-specific IgM responses. These IgM responses are efficient at eliminating parasites expressing their cognate VSG but exert no effect on parasites expressing a different VSG coat. It is possible that when parasites colonize solid tissues, they adapt VSG gene regulation to fine tune antigenic variation. If for example, VSG switching occurred more frequently, perhaps the tissues could be the source of new VSG variants that escape more efficiently the immune response. This hypothesis remains to be investigated.

Another immune evasion strategy employed by *T. brucei* is its ability to remove VSG-bound antibodies through endocytosis of the VSG-Ig complex. This process requires a hydrodynamic drag force to propel the VSG-Ig complex towards the parasite's flagellar pocket and protects the parasite from Ig-mediated opsonization until a critical antibody threshold is reached[51]. Factors affecting this hydrodynamic drag force include the milieu's viscosity, the parasites swimming speed and the interstitial fluid's flow speed[51, 57]. It is possible that these and other biophysical factors may vary between the blood and interstitia of different organs [57], thus granting the parasite a varying capacity to neutralize antibody responses. This could confer to parasites an immune-independent resistance to antibody-mediated phagocytosis.

In summary, we show that an active immune response is mounted in the AT against *T. brucei*, where parasite burden is controlled using the same overarching components of the systemic immune response. Whether these host-protective components are as effective in the AT as in other tissues and the blood requires further study.

## Methods

### Ethics statement

All experimental animal work was performed in accordance with the Federation of European Laboratory Animal Science Associations (FELASA) guidelines and was approved by the Animal Care and Ethical Committee of the Instituto de Medicina Molecular (under the license AWB_2016_07_LF_Tropism).

### RNA isolation and RNA-Seq analysis

Mice infected for 0, 6 and 26 days were euthanized and perfused, gonadal fat depots were collected, RNA extracted by TRIzol (Invitrogen) and its integrity assessed by TapeStation (Agilent). Poly-A mRNA library was prepared as recommended by the manufacturer (Illumina TruSeq) and the samples were sequenced in Illumina HiSeq2000 platform (EMBL and BGI). Sequenced reads were 49bp single-end for samples D0.1, D6.1, D26.1 and D26.2 and 100bp paired-end for the remaining ones. To reduce differences between single and paired-end RNA-Seq datasets, the second read of each mate-pair in paired-end samples was discarded. Also, the first read of each mate-pair was trimmed to the first 49 bases using Trimmomatic (version 0.38)[58].

Read quality was evaluated with FastQC quality control tool (version 0.11.5)[59] and raw reads were trimmed to improve mapping with SolexaQA (version 3.1.7.1)[60]. First, reads were cropped to their longest continuous segment whose PHRED score was higher than 28 and then reads smaller than 25 bp were discarded. Trimmed reads were aligned to the *T. brucei* TREU927 genome (TriTrypDB version 33)[61] using HISAT2 (version 2.0.0-beta)[62] without spliced alignments. Reads mapping to *T. brucei* were discarded and the unmapped reads aligned to the *M. musculus* genome (GRCm38 release 92)[63] using HISAT2 with default parameters. Unique read counts were computed using featureCounts (version 1.6.2)[64] and lowly expressed genes were discarded by keeping genes having a minimum of 10 read counts in at least 2 replicates of the same condition (D0, D6 or D26). In total, 19,488 genes were used to perform differential expression analysis by DESeq2 (version 1.18.1)[65], edgeR (version 3.20.9)[66] and limma (version 3.34.9)[67] from Bioconductor (version 3.5)[68] in the R software environment (version 3.4.4)[69]. Genes having an adjusted *p*-value < 0.01 in at least 2 algorithms and a fold-change > 2 in all were considered differentially expressed.

GO term overrepresentation on differentially expressed genes was performed with the topGO (version 2.30.1)[70] R package, using the weight01 algorithm and Fisher's exact test (p-value < 0.01) for terms with at least 5 annotated genes.

Heat maps were created using the package ComplexHeatmap (version 1.17.1)[71] and ggplot2 (version 3.0.0)[72] was used to create the remaining plots.

Relative immune cell compositions in infected AT were estimated with seq-ImmuCC deconvolution tool[34]. As in differential expression analysis, only genes having a minimum of 10 read counts in at least 2 replicates of the same condition were used. Then, read counts were processed following the authors script in Github[34]: the read counts of each V, D and J gene segments in both T and B cell receptors merged and a quantile normalization performed. The quantile normalized read counts were uploaded in ImmuCC server using the SVR algorithm.

RNA-Seq data from the samples of day 6 post-infection AT were made publicly available in the ArrayExpress database under the accession number E-MTAB-4061. The remaining RNA-- Seq sequence data have also been submitted to the ArrayExpress database under the accession number E-MTAB-7596.

## Animals

*In vivo* experiments were performed with male C57BL/6J mice, from Charles River Laboratories International, unless otherwise stated. *Rag2*-deficient (*Rag2*$^{-/-}$) and *Jht*-deficient (*Jht*$^{-/-}$) mice, generated on a C57BL/6J background, were obtained from Instituto Gulbenkian de Ciência (IGC, Portugal). Ifng-deficient (*Ifng*$^{-/-}$), generated on a C57BL/6J background, were kindly provided by Bruno Silva-Santos laboratory from Instituto de Medicina Molecular (iMM, Portugal). C3-deficient (*C3*$^{-/-}$), generated on a C57BL/6J background, were kindly provided by Miguel Prudêncio laboratory from Instituto de Medicina Molecular (iMM, Portugal). All experimental mice were 7–9 weeks old, unless otherwise stated. *Jht*-deficient mice were 9–21 weeks old, as well as the WT controls in such experiment. Mice were housed in a Specific-Pathogen-Free barrier facility, at iMM, under standard laboratory conditions: 21 to 22˚C ambient temperature and a 12h light/12h dark cycle. Chow and water were available *ad libitum*.

## Parasite lines

Experiments were performed using parasites derived from *T. brucei* AnTat 1.1E, a pleomorphic clone derived from the EATRO1125 strain. AnTat 1.1E 90–13 is a transgenic cell-line encoding the tetracyclin repressor and T7 RNA polymerase[73]. AnTat1.1E 90–13 GFP:: PAD1$_{utr}$ derives from AnTat1.1E 90–13 in which the green fluorescent protein (GFP) is coupled to PAD1 3'UTR.

## Infection

*T. brucei* cryostabilates were thawed and parasite viability by its motility was confirmed under an optic microscope. Mice were infected by intraperitoneal (i.p.) injection of 2,000 *T. brucei* parasites. At selected time-points post-infection, animals were euthanized by $CO_2$ narcosis and immediately perfused transcardially with pre-warmed heparinised saline (50mL phosphate buffered saline (PBS) with 250 μL of 5000 I.U./mL heparin). Organs were collected and either snap frozen in liquid nitrogen; used immediately to prepare single cell suspensions for flow cytometry staining; or immersion-fixed in formalin or glutaraldehyde for histopathology or electron microscopy, respectively. To block PD-1/PD-L1 axis, mice were injected intravenously (i.v.) with 300μg of αPD-1 (clone RMP1-14, InVivoMab, BioXcell), immediately before infection and at days 3, 5 and 7 post-infection.

## Histopathology and Electron Microscopy

Formalin-fixed gonadal adipose tissue was paraffin-embedded and sectioned at 4 μm. Immunohistochemistry for the identification of trypanosomes and inflammatory cells (macrophages and T cells) was performed using a non-purified rabbit serum anti-*T. brucei* VSG13 antigen (cross-reactive with most *T. brucei* VSGs because it is not CRD-depleted, produced in-house), anti-F4/80 antibodies (Abcam, ab6640), anti-CD3 (Dako, A0452), and anti-CD138 (BD, 553712) following conventional protocols. Briefly, antigen retrieval slides was performed in PT Link module (DAKO) at low-Ph, followed by incubation with the primary antibodies. EnVision Link horseradish peroxidase/DAB visualization system (DAKO) was used and

counterstained with Harris hematoxylin. For Transmission Electron Microscopy, samples were fixed with a solution containing 2.5% glutaraldehyde (Electron Microscopy Sciences, EMS) plus 0.1% formaldehyde (Thermo Fisher) in 0.1 M cacodylate buffer (Sigma), pH7.3 for 1 h. After fixation, these were washed and treated with 0.1% Millipore filtered cacodylate buffered (Sigma), post-fixed with 1% Millipore-filtered osmium tetroxide (EMS) for 30 min, and stained *en bloc*, with 1% Millipore-filtered uranyl acetate (Agar Scientific). Samples were dehydrated in increasing concentrations of ethanol, infiltrated and embedded in EMBed-812 medium (EMS). Polymerization was performed at 60°C for 2 days, and ultrathin sections were cut in a Reichert supernova microtome, stained with uranyl acetate and lead citrate (Sigma) and examined in a H-7650 transmission electron microscope (Hitachi) at an accelerating voltage of 100 kV. Electron micrographs were obtained using a XR41M bottom mount AMT digital camera (Advanced Microscopy Techniques Corp). Acquisition of immunohistochemistry images was done in a Nanozoomer-SQ (Hamamatsu Photonics) and analysed using NDP. view2 (Hamamatsu Photonics).

## Parasite quantification in blood and organs

For parasitemia quantification, blood samples were taken daily from the tail vein and diluted 1:150. Parasites were counted manually in a Neubauer haemocytometer (0.1 mm$^3$, detection limit is $3.75 \times 10^5$ parasites per mL of blood). When applicable, the total number of parasites was determined by multiplying by the total volume of blood, considering that a mouse has 58.5 mL of blood per kg of bodyweight.

For parasite quantification in organs, genomic DNA (gDNA) was extracted using NZY tissue gDNA isolation kit (NZYTech, Portugal). The amount of *T. brucei* 18S rDNA was measured by quantitative PCR (qPCR), using the primers 5'-ACGGAATGGCACCACAAGAC–3' and 5'–GTCCGTTGACGGAATCAACC–3', and converted into number of parasites using a calibration curve, as previously described by Trindade *et al.* [15]. Number of parasites per mg of organ (parasite density) was calculated by dividing the number of parasites by the mass of organ used for qPCR. The total amount of parasites in the organ was estimated by multiplying parasite density by the total mass of the organ. Parasite burden normalized to tissue mass and corresponding organ masses are available in the S3 and S5 Files.

## Preparation of single cell suspensions

Gonadal AT samples were incubated at 37°C in Dulbecco's Modified Eagle Medium (DMEM, GIBCO) with Collagenase I (0.4mg/mL, Whortington LS004196), Collagenase IV (1mg/mL, Whortington LS004188) and DNAse (10μg/mL) for 30 minutes, under 1100 rpm agitation. Single cell suspensions from the spleen, heart and digested gonadal AT were obtained by sieving them through a 40μm-pore-size nylon cell strainer (BD Biosciences) with a syringe plunger. Spleen cells were treated with erythrocyte lysis buffer (BioLegend 420301) to lyse red blood cells. Both spleen and gonadal AT cells were resuspended in complete Roswell Park Memorial Institute medium (cRPMI, RPMI supplemented with 1% sodium pyruvate 100mM, 1% MEM non-essential amino acids, 1% HEPES 1M, 1% Pen-Strep, 0.1% gentamycin 50mg/ mL, 0.1% β-mercaptoethanol 50mM and 10% fetal calf serum (all from Gibco)) to use for flow cytometry analysis. Live cells in single cell suspensions were counted after trypan blue staining in a haemocytometer.

## Flow cytometry

Stainings of myeloid and lymphoid cells were performed separately. In isolated spleen cells, stainings were performed in $5 \times 10^6$ cells. Stainings of cells isolated from the gonadal AT were

performed with the highest number of cells possible, by diving the cell suspension equally between each staining (never exceeding $5x10^6$ cells).

For myeloid staining of surface determinants, cells were incubated for 45 minutes at 25˚C in cRPMI, in the presence of 5% normal mouse serum (NMS), with the following antibodies: F4/80-FITC (clone BM8, BioLegend), Ly6G-PerCP/Cy5.5 (clone 1A8, BioLegend), CD274-PE/Cy7 (PD-L1, clone 10F.9G2, BioLegend), CD11b-APC/Cy7 (clone M1/70, BioLegend), CD45-Brilliant Violet (BV) 510 (clone 30-F11, BioLegend) and Ly6C-BV605 (clone HK1.4, BioLegend). To stain non-viable cells, Zombie Violet Fixable Viability Kit (BioLegend) was used, incubating cells in PBS with 5% NMS, for 15 minutes at 25˚C. Finally, stained cells were resuspended in cRPMI for flow cytometry acquisition.

For lymphoid staining, cells were first incubated in cRPMI with phorbol 12-myristate 13-acetate (20ng/mL, PMA) and ionomycin (1µg/mL) to stimulate cytokine production by T cells, for 1h45 at 37˚C. Next, to block cytokine secretion, brefeldin A (10µg/mL) and monensin (5µM), were added for the final 1h at 37˚C of incubation. Staining of surface determinants was performed by incubating cells for 45 minutes at 25˚C in cRPMI with 5% NMS and the following antibodies: CD3-FITC (clone 17A2, BioLegend), CD279-PE (PD-1, clone J43, eBioscience), CD45-BV510 (clone 30-F11, BioLegend), CD4-BV605 (clone RM4-5, BioLegend) and CD8-BV711 (clone 53–6.7, BioLegend). To stain non-viable cells, LIVE/DEAD Fixable Near-IR Dead Cell Stain Kit (Invitrogen) was used, incubating cells in PBS with 5% NMS, for 15 minutes at 4˚C. For staining of intracellular antigens, fixation and permeabilization of cells was performed using Foxp3/Transcription Factor Staining Buffer Set (eBioscience) and the following antibodies: IFN-γ-PerCP/Cy5.5 (clone XMG1.2, BioLegend), Foxp3-APC (clone FJK-16s, eBioscience) and TNF-α-eFluor450 (clone MP6-XT22, eBioscience). Finally, stained cells were resuspended in cRPMI for flow cytometry acquisition.

For staining of B cell surface determinants, cells were incubated for 45 minutes at 25˚C in cRPMI, in the presence of 5% normal mouse serum (NMS), with the following antibodies: CD45-BV510 (clone 30-F11, BioLegend), IgD-APC (clone 12-26c, eBioscience) and CD19-APC/Cy7 (clone 6D5, Biolegend). For staining of intracellular antigens, fixation and permeabilization of cells was performed using Foxp3/Transcription Factor Staining Buffer Set (eBioscience) and the following antibody Ki67-PE (clone 16A8, eBioscience). Samples were analysed on a BD LSRFortessa flow cytometer with FACSDiva 6.2 Software. All data were analysed using FlowJo software version 10.0.7r2. A schematic of the gating strategy used is represented in S2 Fig. Immune cell populations normalized to tissue mass and corresponding organ masses are available in the S3 and S4 Files.

## Soluble VSG isolation and identification

Approximately $5x10^8$ AnTat1.1E parasites cultured in HMI-11 were centrifuged at 2500g at 4˚C for 10 minutes and washed twice in trypanosome dilution buffer (TDB). Trypanosomes were then resuspended and incubated for 3 minutes at 37˚C in 3 mL of 10mM sodium phosphate ($NaH_2PO_4$) buffer pH 8.0 containing a protease inhibitor cocktail (P8340, Sigma). Cells were then incubated for 3 minutes on an ice water bath, centrifuged at 4000g for 5 minutes and the supernatant purified through a diethylaminoethyl Sepharose column (GE Healthcare). Protein quantification was performed using a BCA Protein Assay Kit (A53225, ThermoFisher Scientific) according to the manufacturer's instructions.

The protein sample was run through SDS-PAGE, stained with Coomassie blue and the protein band of interest was isolated, destained, reduced, alkylated and digested with trypsin (Promega) overnight at 37˚C. The tryptic peptides were desalted and concentrated using POROS C18 (Empore, 3M) and eluted directly onto the MALDI plate using 1 µL of in 50% (v/v)

acetonitrile and 5% (v/v) formic acid. The data were acquired in positive reflector MS and MS/MS modes using a 5800 MALDI-TOF/TOF (AB Sciex) mass spectrometer and using TOF/TOF Series Explorer software v.4.1.0 (Applied Biosystems). External calibration was performed using a CalMix5 standard (Protea). The 25 most intense precursor ions from the MS spectra were selected for MS/MS analysis. The raw MS and MS/MS data were analysed using Protein Pilot software v.4.5 (ABSciex) with the Mascot search engine (MOWSE algorithm). For the search parameters were as follows: monoisotopic peptide mass values were considered, maximum precursor mass tolerance (MS) of 50 ppm and a maximum fragment mass tolerance (MS/MS) of 0.3 Da. The search was performed against the SwissProt protein sequence database without taxonomy restriction. Carboxyamidomethylation of cysteines was set as fixed modifications, oxidation of methionines and N-Pyro Glu of the N-terminal Q were set as variable modifications. Protein identification was accepted only when significant protein homology scores were obtained (P < 0.05) and at least one peptide was fragmented with a significant individual ion score (P < 0.05). The MS data were generated by the Mass Spectrometry Unit (UniMS), ITQB/iBET, Oeiras, Portugal.

## Antibody quantification

Antigen-specific antibody titers were determined by ELISA. Assay plates (423501, BioLegend) were coated overnight with 2 μg/mL purified VSG AnTat1.1 (10 mM sodium phosphate buffer, pH 8.0) at 4˚C. Assay plates were blocked with blocking buffer (3% BSA in PBS with 0.05% Tween20). Secondary antibody solutions were prepared by diluting 1:1000 anti-mouse IgM-HRP (lab0372, Covalab), anti-mouse IgG-HRP (lab0365, Covalab) and anti-mouse IgA-biotin (clone RMA-1, BioLegend). Wells with biotinylated antibodies were incubated with SAv-HRP (BioLegend). Assay plates were developed with TMB substrate set (BioLegend), subsequently stopped with a 1M sulfuric acid solution and the OD 450 nm values were read in a TECAN Infinite M200 microplate reader using 570 nm as the reference wavelength. All washing steps were performed with PBS with 0.05% Tween20. The OD cut-off value used for antibody titer was determined according to the formula [Cut-off = $X_{neg}$+ 0.13($X_{pos}$)]sssss, where $X_{neg}$ is the average of the assay wells loaded with PBS and $X_{pos}$ is the average of the wells loaded with the most concentrated samples from mice infected for 9 days Antibody titers for AT, kidney and spleen were normalized to the organ masses used to prepare cell-free suspensions.

## Statistical analysis

The values presented are mean ± SEM. Parasite and immune cell numbers were transformed into their respective Log base 10 values to achieve linearization prior to statistical analysis. Statistical differences were assessed using two-way ANOVA and one-way ANOVA with Sidak's test for multiple comparisons. P values lower than 0.05 were considered to be statistically significant.

## Supporting information

**S1 Fig. Immune activatory and suppressive transcriptomic signature.** Heat map of the differential expression of genes associated with immune response **(A)** activation and **(B)** suppression at days 6 and 26 post-infection relative to non-infected. Gene expression change in Log2 units is denoted in red for up-regulation and in blue for down-regulation.
(TIF)

**S2 Fig. Gating strategy for flow cytometry analysis. (A)** Myeloid gating strategy: live immune cells were gated based on positive expression of CD45 and absence of viability dye

signal. Single cells were then identified using SSC-W vs FSC-A gating. Total myeloid cells were selected based on CD11b expression and then subdivided into neutrophils and other myeloid cells based on Ly6G gating. Within the remaining myeloid cells, macrophages and monocytes were identified using F4/80 vs Ly6C gating. (**B**) Lymphoid gating strategy: single live immune cells were identified as described above and then T cells were identified based on the co-expression of CD3 and CD4 or CD3 and CD8. CD4+ T cells were further subdivided into conventional CD4+ T cells or regulatory T cells based on FoxP3 expression. Effector T cells were identified by gating single or dual expression of TNF-α and IFN-γ within CD3+CD4+FoxP3- or CD3+CD8+ cells (**C**) B cell gating strategy: Live immune cells were identified as before and then B cells were identified based on expression of CD19 and lack of CD3 expression. Activated B cells were identified based on positive Ki67 expression and absence of IgD expression. (TIF)

**S3 Fig. Comparison of flow cytometry and immuCC data.** Variation of immune cell subsets between days 0, 6 and 26 post-infection. Data are scaled between 0 and 1, where 1 corresponds to the highest percentual value within each group. (**A**) B cells, (**B**) CD4+ T cells, (**C**) CD8+ T cells, (**D**) Macrophages, (**E**) monocytes, (**F**) neutrophils and (**G**) other immune cells. (TIF)

**S4 Fig. Organ mass and T cell dynamics.** (**A**) Spleen and gonadal AT mass. (**B**) CD4+ T cells. (**C**) CD8+ T cells. Error bars represent the standard error of the mean (n = 2–6 mice per group). Statistical analysis was performed with a two-way ANOVA using Sidak's test for multiple comparisons. * refers to statistical differences between the group in each time-point and the non-infected group. *, $P<0.05$; **, $P<0.01$; ***, $P<0.001$; ****, $P<0.0001$. (TIF)

**S5 Fig. Purified soluble VSG identification.** (**A**) Coomassie stained SDS-PAGE of purified soluble VSG solution denoting a single preeminent band within the predicted size range of VSG. (**B**) Best peptide match within the SwissProt database (sp|P06015|VSA1_TRYBB, Variant surface glycoprotein AnTaT 1.1 OS = Trypanosoma brucei brucei OX = 5702 PE = 2 SV = 1) with matching peptides depicted in bold red. (**C**) Summary of protein identification. Protein score is -10*Log(P), where P is the probability that the observed match is a random event. Protein scores greater than 70 are significant ($p<0.05$). Protein scores are derived from ions scores as a non-probabilistic basis for ranking protein hits. (**D**) Measurement of anti-VSG IgM titers by ELISA depicting optical density (OD) curves and antibody titer cut off determination. (TIF)

**S6 Fig. Assessment of the role of PD-1 expression in parasite control.** Percentage of PD1+ (**A**) CD4+ T cells and (**B**) CD8+ T cells. Effect of anti-PD-1 treatment on (**C**) parasitemia of mice infected with *T. brucei*, quantified in a hemocytometer and (**D**) number of T. brucei parasites quantified by qPCR in spleen, gonadal AT and heart, 6 and 9 days post-infection. Error bars represent the SEM (n = 5 mice per group). Statistical analysis was performed with a two-way ANOVA using Sidak's test for multiple comparisons. (**A-B**) * refers to statistical differences between groups. *, $P<0.05$; **, $P<0.01$; ***, $P<0.001$; ****, $P<0.0001$. (TIF)

**S1 Table. Mapping information of RNA-Seq reads in samples from infected AT.** (DOCX)

**S1 File. RNA-seq raw counts and differential expression analysis.** (**A**) Raw counts and (**B**) reads per kilobase per million mapped reads from RNA-seq of AT from non-infected mice

(D0), n = 3, day 6 post-infection (D6), n = 3 and day 26 post-infection (D26), n = 2. Differential expression analysis between **(C)** D0 and D6, **(D)** D0 and D26 and **(E)** D6 and D26.
(XLSX)

**S2 File. Significant GO terms list.** GO term analysis from RNA-seq of AT from non-infected mice (D0), n = 3, day 6 post-infection (D6), n = 3 and day 26 post-infection (D26), Up-regulated GO terms in the AT between **(A)** D6vsD0, **(B)** D26vsD0, **(C)** D26vsD6. Down-regulated GO terms in the AT between **(D)** D6vsD0, **(E)** D26vsD0, **(F)** D26vsD6.
(XLSX)

**S3 File. Effect of tissue weight on immune cell number and parasite number determination. (A)** Weight of spleens and ATs analysed. Data for entire organs and normalized to tissue weight **(B-C)** parasite burden, **(D-E)** CD45+ cells, **(F-G)** neutrophils, **(H-I)** macrophages, **(J-K)** monocytes, **(L-M)** effector CD4+ T cells, **(N-O)** effector CD8+ T cells and **(P-Q)** regulatory T cells.
(PDF)

**S4 File. Effect of tissue weight on B cell number determination. (A)** Weight of spleens and ATs analysed. Data for entire organs and normalized to tissue weight **(B-C)** B cells and **(D-E)** activated B cells.
(PDF)

**S5 File. Effect of tissue weight on parasite number determination.** Weight of organs used from infected **(A)** $Rag2^{-/-}$, **(B)** $Jht^{-/-}$, **(C)** $Ifng^{-/-}$, **(D)** $C3^{-/-}$ and **(E)** anti-PD1 treated mice, with respective controls. Data for parasite numbers in the entire organ or normalized to tissue weight for infected **(F-G)** $Rag2^{-/-}$, **(H-I)** $Jht^{-/-}$, **(J-K)** $Ifng^{-/-}$, **(L-M)** $C3^{-/-}$ and **(N-O)** anti-PD1 treated mice, with respective controls.
(PDF)

## Acknowledgments

We are grateful to Bruno Silva-Santos (iMM) for the helpful discussion and for providing experimental materials. We thank Andreia Pinto, Ana Margarida Biscaia Santos and Ana Rita Pires from the Histology and Comparative Pathology Laboratory of the iMM for expert technical assistance. We also thank the staff of the Rodent facility of the iMM and Catarina Correia and Isabel Abreu of the Mass Spectrometry Unit (UniMS) of ITQB/iBET.

## Author Contributions

**Conceptualization:** Tiago Bizarra-Rebelo, Karine Serre, Luisa M. Figueiredo.

**Data curation:** Henrique Machado, Tiago Bizarra-Rebelo, Mariana Costa-Sequeira, Barbara Rentroia-Pacheco.

**Formal analysis:** Henrique Machado, Tiago Bizarra-Rebelo, Mariana Costa-Sequeira, Barbara Rentroia-Pacheco.

**Funding acquisition:** Karine Serre, Luisa M. Figueiredo.

**Investigation:** Henrique Machado, Tiago Bizarra-Rebelo, Sandra Trindade, Tânia Carvalho, Filipa Rijo-Ferreira, Karine Serre.

**Methodology:** Tiago Bizarra-Rebelo, Mariana Costa-Sequeira, Tânia Carvalho, Barbara Rentroia-Pacheco.

**Project administration:** Karine Serre, Luisa M. Figueiredo.

**Resources:** Karine Serre, Luisa M. Figueiredo.

**Software:** Mariana Costa-Sequeira, Barbara Rentroia-Pacheco.

**Supervision:** Karine Serre, Luisa M. Figueiredo.

**Validation:** Henrique Machado, Tiago Bizarra-Rebelo, Karine Serre, Luisa M. Figueiredo.

**Visualization:** Henrique Machado, Tiago Bizarra-Rebelo, Mariana Costa-Sequeira.

**Writing – original draft:** Henrique Machado, Tiago Bizarra-Rebelo, Tânia Carvalho, Karine Serre, Luisa M. Figueiredo.

**Writing – review & editing:** Henrique Machado, Tiago Bizarra-Rebelo, Tânia Carvalho, Karine Serre, Luisa M. Figueiredo.

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
