## [Decision Letter · Decision Letter 0]

10 Jun 2021

Dear Authors,

Thank you very much for submitting your manuscript "Trypanosoma brucei  triggers a broad immune response in the adipose tissue" for consideration at PLOS Pathogens. As with all papers reviewed by the journal, your manuscript was reviewed by members of the editorial board and by several independent reviewers. In light of the reviews (below this email), we would like to invite the resubmission of a significantly-revised version that takes into account the reviewers' comments.

As indicated by all three reviewers, the manuscript is well written and covers a unique field of trypanosomosis research.

However, the reviewers also flag that the there are some issue with the 'depth' of the analysis, in particular when it comes to the B-cell/antibody data, and the transcriptomic analysis. Addressing these concerns would improve the quality of the paper, and would have an added value in terms of future referencing of the study.

We cannot make any decision about publication until we have seen the revised manuscript and your response to the reviewers' comments. Your revised manuscript is also likely to be sent to reviewers for further evaluation.

Sincerely,

Stefan Magez

Guest Editor

PLOS Pathogens

David Sacks

Section Editor

PLOS Pathogens

Kasturi Haldar

Editor-in-Chief

PLOS Pathogens

orcid.org/0000-0001-5065-158X

Michael Malim

Editor-in-Chief

PLOS Pathogens

orcid.org/0000-0002-7699-2064

Reviewer's Responses to Questions

**Part I - Summary**

Reviewer #1: Figueiredo et al had previously published the seminal finding that trypanosomes can interact with adipose tissue metabolically. This finding raised the possibility that trypanosomes might can "hide" from the immune response during their AT resident phase, thus evading clearance. The current manuscript reports on experiments meant to address this possibility.

First, they use transcriptomic analyses of infected tissue to demonstrate the increase in inflammatory signatures. Second, they characterize the type of infiltrating immune cell by FACS. These experiments are by necessity observational, but also demonstrate clearly that tissue responds to infection - a neat finding.

The rest of the paper is less strong. The response to T.brucei in blood is clearly predicated on the presence of IgM (or IgD) antibodies. Consequently the authors look for B cells in AT, but not for the CD138+TACI+ plasma cells that would secrete those antibodies. Without such data it is entirely plausible that Ig within AT is just spilling out from the blood (where the levels are of course highest). Also, though it is entirely plausible that the antibodies in question are antigen (ie VSG) specific, as far as I could tell antibody levels here are reported as "total" IgM or "total" IgG or IgA (ie not antigen specific - which is the only relevant compound here).

Reviewer #2: The manuscript of the Figueiredo group for the first time explores the immunological responses in Trypanosoma brucei infected adipose tissue to evaluate whether this tissue can be considered to provide some sort of immunological sanctuary. The authors clearly illustrate that several inflammatory genes are upregulated upon infection of the gonadal adipose tissue, and with infiltration of CD45+ cells, consisting of myeloid (monocytes, macrophages and neutrophils) and lymphoid cells (effector and regulatory CD4+ T cells, CD8+ T cells and B cells). Various antibody isotypes (IgM, IgG and IgA) are shown to be increased in the gonadal adipose tissue. Using three different knockout mouse models (Rag2-/-, Jht-/- and Ifng-/-), authors document that T and B cells as well as IFN-g contribute to parasite clearance in adipose tissue between days 6 and 9 post infection, mirroring the effector mechanisms responsible for reduction of blood parasitemia within that particular window of infection. The manuscript is well written and the data clearly presented. A number of comments are included below to be considered for a revised version of the manuscript.

Reviewer #3: This is a well written paper and informative and novel. It expands on the discovery of adipose tissue as a niche for extracellular African trypanosome residency and explores the immune response therin.

The route of infection is classically used in the trypanosome field and thus relevant, but it does leave one wondering about the response that would be seen in response to a tsetse fly bite or an intradermal infection model as a proxy. I do hope this wonderful team of scientists will investigate this in the future.

**Part II – Major Issues: Key Experiments Required for Acceptance**

Reviewer #1: Additional Comments:

1) I would suggest that they use a tool like TIMER2.0 to attempt to deconvolute the bulk RNAseq data into a model of the types of infiltrating cells (as a cross-check of the FACS data).

2) Lines 197-199 ("We performed IHC for parasites using an anti-VSG antibody... in sections...on days 6 and 26 post infection"). Is this a pan-VSG antibody? (Does such a thing exist?) Would a specific VSG serotype present in fat at day 6 remain there till day 26? This needs to be clarified.

3) Lines 182-183 ("immune cells are recruited .. in AT .. increasing steadily as infection progresses"). Does this not suggest an inability of the immune response to clear parasites? or is it simply an increase proportional to the number of parasites in that tissue as a function of time?

Reviewer #2: Major comments, which may be potentially resolved without additional experimental work but require additional analyses and modification of the manuscript:

Did the authors include fluorescence minus one controls in the flow cytometry analyses to support the gating strategy and presented data. Especially for the staining of the various T cell subsets, this would need to be included.

The transcript data are presented and discussed rather cryptically, with a hierarchical clustering, GO term analysis and only a single supplementary figure describing the Th1 signature genes. A gradual Treg infiltration is observed over the course of infection by flow cytometry, but no connection is made with the transcriptome data. Major gene transcripts for genes such as il10, tgfb, nos2 are not discussed. In general, one would expect a much more in depth analysis on differential pathways, and not only limited to the upregulated transcripts.

In a previous study by the research group, transcript levels of the active variant surface glycoprotein were found to be 3-fold lower in adipose tissue than in blood. Can the authors comment on how this can be reconciled with higher overall parasite burdens despite the immune surveillance and parasite elimination in the AT. From the presented analyses, it seems that it remains to be understood why the AT accumulates relatively higher parasite loads. Do the authors have a prevailing hypothesis?

Reviewer #3: Figure 1 Page 7- The transcriptome profile was generated using non-infected mice as controls, which discounts the inflammatory response due to the injection that is done in peritoneal cavity, which is in close proximity to the adipose tissue. Therefore, a mock injected mouse is the proper control for this figure.

That being said the response to bacteria evolves with time D6 vs D26, are the parasites or immune cells disrupting the intestinal mucosa (epithelial barrier)? Why is this response present?

Figure 3. What adipose tissue is being usd in this figure? I looked up perirenal AT (3B) and it is surrounding the kidney, hence peri "renal". Why was this used and not the gonadal AT as outlined in the methods section pg 25? This necessitates clarity for all the AT samples used in this paper and WHY.

Figure 4. The result that innate immune cells such as neutrophils and particularly macrophages infiltration is prominent especially during the low parasite count is very interesting. However, the authors ignore the function of macrophages and focus on B and T cell function. It would be good see how depletion of macrophages affects parasite survival in the mouse using mouse models such as CD11b-DTR. Has this been done by others in the field? if so please discuss.

Figure 4 and 5- How does immune cell infiltration and humoral immunity in mock injected mouse (Control) look?

In there any data in the literature to address this question?

Figure 5. Where are the macrophages? Was this data collected? If so please share, opsonization is a major clearance mechanism of parasites. This has to be relevant in the overall outcome. Yes I know this the humoral immunity figure.

Page 18 line 355- it should be systemically not systematically.

Page 17, Lines 348-350- The data is not sufficient to make the statement as parasites in AT is similar to blood which could be due to residual blood in the tissues.

The paper is novel in that it studies the immunity in the solid organ such as adipose tissue where the parasites could potentially hide. However, the occurrence of Trypanosoma brucei as well as immune cells is studied only in intraperitoneal infection model, which is not the natural route of infection. It is therefore essential to mention in the manuscript that it is not the normal route of infection and hence the pathogenesis and tissue tropism of disease may be different in the natural infection.

**Part III – Minor Issues: Editorial and Data Presentation Modifications**

Reviewer #1: (No Response)

Reviewer #2: Minor comments:

The introduction is very succinct and would benefit some extension to introduce the various immune parameters analyzed in the manuscript that are known to be relevant for T. brucei control.

Please specify the hematocytometer used

Was the antibody quantification done on the cell free supernates of single cell suspension prepared following the collagenase/DNAse treatment as for the flow cytometry?

Reviewer #3: Page 3 lines 59-62- Rewrite sentence for clarity.

These data show that parasite

349 elimination in AT, from days 6 to 9 post-infection, relies (this is a strong word.....please tone down) on B and T cell responses,

350 similarly to parasitemia clearance in the blood.

Please give the total mass of the organ for each mouse in each experiment in the text/graph/figure legend, so we can understand if there was expansion of the organ or not. AND make a another graph with number of cells or parasites per mg of organ.

Number of parasites per mg of organ (parasite density) was

568 calculated by dividing the number of parasites by the mass of organ used for qPCR. The

569 total amount of parasites in the organ was estimated by multiplying parasite density by

570 the total mass of the organ.

What was a lower isolation? why and how was this determined?

In isolated gonadal AT cells, due to

589 a lower isolation yield,

Another immune evasion strategy employed by T. brucei is its ability to remove

440 VSG-bound antibodies through endocytosis of the VSG-Ig complex. This process

441 requires a hydrodynamic drag force to propel the VSG-Ig complex towards the parasite’s

442 flagellar pocket and protects the parasite from Ig-mediated opsonisation until a critical

443 antibody threshold is reached[41]. Factors affecting this hydrodynamic drag force include

444 the milieu’s viscosity, the parasites swimming speed and the interstitial fluid’s flow

445 speed[41]. It is possible that these factors may vary between the interstitia of different

446 organs, thus granting the parasite a varying capacity to neutralize antibody responses.

447 This could confer to parasites an immune-independent resistance to antibody-mediated

448 phagocytosis. SURELY THE DIFFERENCE WOULD BE MOST MARKED BETWEEN BLOOD AND INTERSTITIAL FLUID...PLEASE COMMENT

PLOS authors have the option to publish the peer review history of their article (what does this mean?). If published, this will include your full peer review and any attached files.

Reviewer #1: **Yes: **F Nina Papavasiliou

Reviewer #2: No

Reviewer #3: No
---

## [Editor Report · Decision Letter 1]

31 Aug 2021

Dear Dr. Figueiredo,

We are pleased to inform you that your manuscript 'Trypanosoma brucei  triggers a broad immune response in the adipose tissue' has been provisionally accepted for publication in PLOS Pathogens.

Best regards,

Stefan Magez

Guest Editor

PLOS Pathogens

David Sacks

Section Editor

PLOS Pathogens

Kasturi Haldar

Editor-in-Chief

PLOS Pathogens

orcid.org/0000-0001-5065-158X

Michael Malim

Editor-in-Chief

PLOS Pathogens

orcid.org/0000-0002-7699-2064

We would like to thank the authors for the very extensive revision of the paper and the work done to try and reply to all the reviewer's questions
---

## [Editor Report · Acceptance letter]

10 Sep 2021

Dear Dr. Figueiredo,

We are delighted to inform you that your manuscript, "*Trypanosoma brucei*  triggers a broad immune response in the adipose tissue," has been formally accepted for publication in PLOS Pathogens.

Best regards,

Kasturi Haldar

Editor-in-Chief

PLOS Pathogens

orcid.org/0000-0001-5065-158X

Michael Malim

Editor-in-Chief

PLOS Pathogens

orcid.org/0000-0002-7699-2064